# A Large Case Series of Neurocysticercosis in Kuwait, a Nonendemic Arabian Gulf Country in the Middle East Region

**DOI:** 10.3390/microorganisms9061221

**Published:** 2021-06-04

**Authors:** Jamshaid Iqbal, Suhail Ahmad, Mohammad Al-Awadhi, Amir Masud, Zainab Mohsin, Abdullah Y. Abdulrasoul, Khalifa Albenwan, Nadia Alenezi, Fatima AlFarsi

**Affiliations:** 1Department of Microbiology, Faculty of Medicine, Kuwait University, P. O. Box 24923, Safat 13110, Kuwait; suhail.ahmad@ku.edu.kw (S.A.); mohammad.alawadhi@grad.ku.edu.kw (M.A.-A.); khalifa.albenwan@ku.edu.kw (K.A.); 2Al-Sabah Hospital, Ministry of Health, Kuwait City 13001, Kuwait; dramirmasud88@hotmail.com; 3Farwaniya Hospital, Ministry of Health, Farwaniya 81004, Kuwait; dr.zainabmohsin@gmail.com; 4Jahra Hospital, Ministry of Health, Alsafat 01753, Kuwait; abdullah_al_mosawi@hotmail.com; 5Parasitology Reference Laboratory, Mubarak Al-Kabir Hospital, Jabriya 47060, Kuwait; nadiaalenezi228@gmail.com (N.A.); laabeebaa.fa87@gmail.com (F.A.)

**Keywords:** neurocysticercosis, enzyme-linked immunotransfer blot (EITB), imported cysticercosis, Kuwait, epidemiology, Middle East region, prevalence

## Abstract

Neurocysticercosis (NCC), a leading global cause of severe progressive headache and epilepsy, in developed or affluent countries is mostly diagnosed among immigrants from poor or developing *Taenia solium* taeniasis-endemic countries. Taeniasis carriers in Kuwait are routinely screened by insensitive stool microscopy. In this study, enzyme-linked immunoelectrotransfer blot (EITB) was used as a confirmatory test for NCC. Screening was performed on 970 patients referred for suspected NCC on the basis of relevant history and/or ring-enhancing lesions on computed tomography and/or magnetic resonance imaging during a 14-year period in Kuwait. Demographic data and clinical details were retrieved from laboratory or hospital records. EITB was positive in 150 subjects (15.5%), including 98 expatriates mostly originating from taeniasis-endemic countries and, surprisingly, 52 Kuwaiti nationals. The clinical details of 48 of 50 NCC cases diagnosed during 2014–2019 were available. Most common symptoms included seizures, persistent headache with/without fever, and fits or loss of consciousness. Cysticercal lesions were located at various brain regions in 39 of 48 patients. Multiple members of 3 families with NCC were identified; infection was linked to domestic workers from taeniasis-endemic countries and confirmed in at least 1 family. Our data show that NCC is predominantly imported in Kuwait by expatriates originating from taeniasis-endemic countries who transmit the infection to Kuwaiti citizens.

## 1. Introduction

The ingestion of larval cysts of *Taenia solium* by human subjects in undercooked pork causes taeniasis, a mild intestinal infection. However, the ingestion of food or drinks contaminated with human fecal material from *T. solium* taeniasis patients or carriers containing parasite eggs leads to the development of cysts in different body tissue types in the recipients (cysticercosis) or to cyst development in the central nervous system (neurocysticercosis) [1,2]. Cestode *T. solium* is responsible for a considerable cross-sectoral health and economic burden due to human NCC and porcine cysticercosis. According to World Health Organization (WHO) estimates, *T. solium* is recognized as a leading cause of death from a foodborne disease that causes the loss of nearly 2.8 million disability-adjusted life-years (DALYs) and is responsible for 2.56–8.30 million global NCC cases [3]. It is estimated that 80% of the world’s 50 million cysticercosis cases live in poor, developing countries. Cysticercosis is highly endemic in Latin American, sub-Saharan Africa, South/Southeast Asia, and the Indian subcontinent, where domestic pig husbandry is practiced, and is estimated to affect between 2.56–8.30 million individuals [4,5,6,7,8,9]. NCC is a debilitating yet preventable neurological manifestation that causes severe progressive headache and up to 70% of preventable epilepsy cases in *T. solium*-endemic countries [1,3,10].

The growing number of immigrants from endemic areas, increased tourism, and international business affairs have rendered people from nonendemic areas more susceptible to acquiring taeniasis and cysticercosis [11]. Human cysticercosis is increasingly recognized in the developed and industrialized nonendemic countries of Western Europe, North America, and Australia likely due to increased pork consumption, immigration, and international travel [7,12,13,14,15,16,17,18,19,20]. On the other hand, taeniasis/cysticercosis is more common in Eastern Europe, where regulatory inspections and oversight are compromised due to various socioeconomic and political developments [21].

*T. solium* taeniasis/cysticercosis is included among the list of 20 neglected tropical diseases identified by the WHO [3]. The control and elimination of cysticercosis, and its affiliated complications are mainly dependent on the early diagnosis of taeniasis. However, the accurate diagnosis of cysticercosis and NCC, based on stool microscopy in most developing countries, is difficult, particularly if neuroimaging findings are not available [1]. A more reliable test for the diagnosis of NCC is based on the detection of *T. solium*-specific antibodies in serum samples from suspected patients. The antibodies are detected by enzyme-linked immunoelectrotransfer blot (EITB) assay. This test has sensitivity of 97.5% and specificity 100% for the diagnosis of NCC in patients with >2 viable cysts in the nervous system [22,23].

Kuwait is an affluent Arabian Gulf country in the Middle East. It has a total population of 4.7 million individuals, which includes only 1.46 million Kuwaiti nationals. The remaining 3.21 million people are expatriate workers or their dependents (https://www.paci.gov.kw/stat/Default.aspx, accessed on 9 March 2021) [24]. Most expatriates, including domestic helpers and food handlers in restaurants, originate from countries such as India, Bangladesh, Nepal, Sri Lanka, Philippines and Ethiopia, which are endemic for *T. solium* taeniasis and other infectious diseases [25,26,27,28,29,30]. Currently, infected individuals in Kuwait are usually screened for taeniasis by stool microscopy for the detection of ova and proglottids. Although the test is rapid, it has very low sensitivity and specificity, particularly if it is performed on a single fecal material [28]. Information on the incidence of taeniasis and/or cysticercosis among Middle East region countries including Kuwait is limited but presumed to be low, as pork consumption, and swine handling and farming are prohibited in most Middle East region countries due to the Islamic culture [28,31]. Nonetheless, sporadic cases of cysticercosis have been reported from Kuwait and other Middle East region countries over the years. Several reports indicate that the prevalence of NCC cases in the Middle East has increased in recent years, with most patients locally acquiring the infection and presenting with a single parenchymal brain cysticercus granuloma with minimal or no apparent symptoms [25,31,32,33,34,35,36].

In this study, we retrospectively determined the prevalence of cysticercosis/NCC during a 14-year period (January 2006 to December 2019) in Kuwait. Serological testing with EITB assay was performed on patients who were referred for suspected NCC on the basis of relevant history, physical examination, and/or ring-enhancing lesions on computed tomography and/or magnetic resonance imaging (CT/MRI) observations. A retrospective review of all cases with a confirmed diagnosis of NCC was performed. Patient sociodemographic and clinical data, and other relevant details were retrieved from the laboratory and hospital records, and corelated with laboratory findings.

## 2. Methods

### 2.1. Patients and Clinical Specimens

A total of 970 patients were referred to the Parasitology Reference Laboratory (PRL), Faculty of Medicine, Kuwait University, Kuwait for suspected NCC on the basis of relevant history, including household close contact with confirmed NCC patients and/or clinical findings of ring-enhancing lesions on CT/MRI in various hospitals across Kuwait during a 14-year period (January 2006 to December 2019). Kuwait is a small country in the Middle East which mainly comprises Kuwait City with many governorates, and two other smaller towns, Jahra and Ahmadi, located within 40–50 km from Kuwait City. Clinical samples were obtained from all major government hospitals spread out within a radius of ~40 km across Kuwait [37,38]. Clinical specimens (blood and/or cerebrospinal fluid, CSF) were obtained from the suspected cases after obtaining verbal consent only as part of routine patient care and diagnostic work-up. Sera from clotted blood samples were separated, and serum or CSF was used as the source of anticysticercal IgG antibodies. All suspected NCC cases were referred to the Parasitology Reference Laboratory at the Faculty of Medicine for conformation of NCC with EITB assay.

The study was approved by the Ethical Committee of the Health Sciences Center, Kuwait University, and the Ethical Committee for the Protection of Human Subjects in Research, Ministry of Health, Kuwait (reference no. 187/2014); the need for informed consent was waived, as no direct contact with the patients was made for this specific study. Personal details about individual participants were kept confidential, data were anonymously analyzed, and the results on deidentified samples are reported in this study.

### 2.2. Cysticercal Immunoblot for Anticysticercal IgG Antibodies

EITB assay was performed by using the QualiCode™ Cysticercosis Immunoblot Kit (Immunetics Inc., Boston, MA, USA), and the procedure and interpretation of results were carried out according to the manufacturer’s instructions. Briefly, antigen-bound strips were placed in 1 mL of 1X wash buffer for 1 min in the channels of the incubation tray. The wash buffer was aspirated, and 1 mL of sample diluent and 10 µL of test serum samples, and positive and negative controls included in the kit were added and incubated for 1 h. Samples were aspirated, and strips were washed twice with 1 mL of 1X wash buffer for 3 min. Subsequently, 1 mL of conjugate (antihuman IgG antibody) was added to each channel, and strips were incubated for 1 h. After washing the strips twice with 1 mL of 1X wash buffer and with 1 mL distilled water, 1 mL of alkaline phosphatase substrate solution was added to each channel, and the strips were incubated for 8 min. Contents were removed, strips were washed twice with distilled water and air-dried, and results were interpreted. Bands on the strips were identified with a reference card, and positive control strips showed bands corresponding to p50 and p14. Positive serum samples were identified by bands corresponding to p50, p42–39, p24, p21, p18, and/or p14. The presence of a minimum of 2 well-defined bands (p50 and p14) among the 5 described bands was indicative of cysticercosis/NCC.

### 2.3. Extraction of Epidemiological and Clinical Data and Location of Cysticercal Lesions

Epidemiological data for the individuals referred for cysticercosis investigation were retrieved from the laboratory logbooks of PRL and from the digital Laboratory Information System (LIS), and entered into a Microsoft Excel Worksheet (.xlsx) in groups based on age, gender, nationality, referring hospital or region, date of sample collection, and diagnostic result. Individuals whose nationalities were missing in the logbooks were recorded as subjects of ‘unknown nationality’. Clinical details and symptoms at presentation in the hospital associated with NCC were retrieved from medical records of the patient’s files and from the digital Hospital Information System (HIS). The locations of cysticercal lesions were determined on the basis of neuroimaging findings from MRI/CT scans.

### 2.4. Statistical Analysis

Descriptive statistical analysis was performed, and Pearson’s chi-squared test was used to test association of infection with patient’s gender, nationality, location, travel history (if available), and the year of sample collection. The difference in age between infected and uninfected patients was measured by Student’s t-test, and a *p* value < 0.05 was considered to be significant. Data were analyzed by WinPepi software v11.65, as previously described [28].

## 3. Results

A total of 970 subjects suspected of NCC were referred from different hospitals in Kuwait during the 14-year study period (January 2006 and December 2019) to PRL for confirmation by serological screening. On the basis of EITB results, 150 (15.5%) subjects tested positive. Serum samples from most of the subjects showed reactivity with all 7 diagnostic antigens, and all 150 subjects showed reactivity at least with p50 and p14 antigens. The remaining 820 samples showed no reaction with any of the *T. solium*-specific antigens (Table 1).

The demographic details of EITB-positive and -negative subjects are shown in Table 1. Due to the retrospective nature of the study, information regarding the age of patients was not available for 296 subjects whose serum samples were processed during 2006–2009, while the exact nationality of 124 of 970 patients was also not available. Despite this limitation, the average age of cysticercosis cases was significantly lower (24.3 years) than that of serologically negative subjects (32.8 years), while the male-to-female ratio in the two groups was nearly the same (Table 1). The country of origin was known for 130 of 150 cysticercosis cases, and comprised 52 Kuwaiti subjects and 78 expatriates originating from India (*n* = 55), Nepal (*n* = 12), Bangladesh (*n* = 4), Syria (*n* = 3), Pakistan (*n* = 1), Sri Lanka (*n* = 1), the Philippines (*n* = 1), and South Korea (*n* = 1) (Table 1). The country of origin for the remaining 20 non-Kuwaiti subjects was not known (Table 1).

EITB results were also analyzed for positivity among serum samples investigated every year during the 14-year period (2006 to 2019), and the nationality of NCC-positive subjects and the data are presented in Table 2. Data showed that immunoblot positivity for NCC among suspected cases showed a biphasic curve. Positivity remained nearly the same (~20%) during the first 6 years of the study (2006 to 2011), declined sharply in 2012, increased slowly during the next 3 years (2013 to 2015), before declining again to very low levels during the last 3 years. The same pattern was seen among immunoblot-positive Kuwaiti patients except for in 2014, when the maximal number (*n* = 16) of suspected Kuwaiti patients tested positive. However, the pattern among immunoblot-positive Indian patients, the dominant expatriate group, paralleled the overall positivity rate (Table 2). The EITB was also performed on 49 CSF samples, but it was positive only from one Indian patient in 2018, who also had a positive result from the blood sample. We also detected NCC cases among multiple households of at least 3 families who had domestic workers from taeniasis-endemic countries working in their homes. The infection was confirmed among 2 domestic workers from Sri Lanka and the Philippines who were working for a Kuwaiti family, and 4 NCC cases were detected among the residents in this household (see below).

Due to the retrospective nature of the study and the detection of immunoblot-positivity among 98 expatriates, many of whom had likely left the country (as they were untraceable), clinical details for only 48 of 50 NCC patients detected during 2014 to 2019 were available and are presented in Table 3. The most common clinical presentation included tonic–clonic seizures (*n* = 36), persistent headache (*n* = 14), persistent headache with fever (*n* = 7), and seizures with loss of consciousness (*n* = 6). Among 36 patients with tonic–clonic seizures, 30, 4, and 2 patients experienced generalized seizures, focal seizures, and status epilepticus, respectively. Furthermore, 10 cases had experienced a single seizure, while 26 cases had had ≥2 seizures over time before clinical samples were taken for serological testing. Only 9 patients presented no apparent symptoms, and they mostly included domestic workers from taeniasis-endemic countries (Table 3).

Cysticercal lesions were observed in CT/MRI images in 39 of 48 (81.2%) NCC cases detected during 2014 to 2019. A total of 15 cases (38.5%) had a single lesion, and 24 (61.5%) NCC cases showed multiple brain lesions at multiple sites. Cerebral edema was observed around 11 cystic lesions. More than 47 cyst locations were identified by neuroimaging in 39 NCC cases; a vast majority (31 of 39, 79.5%) of cases showed parenchymal lesions. Subarachnoid lesions were observed in 3 (7.69%) cases, and the rest were located at multiple locations or sites (Table 4). A total of 23 of 39 (59%) cases showed lesions in their frontal lobes, followed by the occipital (12 cases, 30.7%) and parietal (10 cases, 25.6%) lobes (Table 4). Interestingly, >68% of the frontal-lobe lesions were on the left side of the cerebral hemisphere. Some cases also had cysticercal lesions at unusual sites; at least 3 cases showed multiple lesions at 4 sites, 2 cases had lesions in the pineal body, and 1 case showed multiple cerebral lesions, in addition to extra-axial CNS cysts in the pterygoid muscle, left parotid gland, and right temporalis muscle.

## 4. Discussion

The increasing frequency of NCC patients in nonendemic countries, including the Middle East, is an important public-health concern that requires vigilant physicians and efficient diagnostic protocols for the timely detection of *T. solium* carriers from endemic countries for preventing infection transmission to the local population [11,18]. To our knowledge, this is the first study reporting on diagnostic and clinical characteristics of a large series of NCC patients in Kuwait and the Middle East.

The diagnosis of cysticercosis and NCC is challenging, as cases generally present with complex variations in clinical manifestations and nonpathognomonic neuroimaging findings in most cases [1,8,39]. The patients suspected of cysticercosis or NCC after initial evaluation on the basis of a careful patient history, physical examination, and neuroimaging studies comprising noncontrast CT scan and/or MRI were referred to the PRL for the confirmation of cysticercosis/NCC by serological methods, as the infected individuals develop a predominantly IgG response due to tissue invasion and cyst formation by the larval parasite [40]. An easier and rapid ELISA, and a more specific and reliable EITB assay (QualiCode™ Cysticercosis Immunoblot Kit, Immunetics Inc., Boston, MA, USA manufactured under license from the U.S. Government) that uses sera obtained from patients with a clinically diagnosed active disease were developed as screening tests for NCC [22,40]. The assay utilizes seven lentil–lectin affinity-purified glycoprotein antigens extracted from raw *T. solium* cysticerci. The anticysticercal IgG antibodies potentially present in the serum sample of cysticercosis patients selectively bind to 1 or more of 7 specific antigens on the strip, and antigen–antibody complexes are revealed as purple bands with alkaline phosphatase–antihuman IgG conjugate. The EITB assay identifies positive cysticercosis or NCC cases with 97.5% sensitivity and 100% specificity in patients with multiple parenchymal lesions with ventricular or subarachnoid NCC [22,23,41,42]. However, sensitivity is lower in patients with a single parenchymal lesion or with only calcifications. The EITB assay was approved as a confirmatory test in patients with suspected NCC according to the guidelines for the clinical management of patients with NCC by the American Society of Tropical Medicine and Hygiene (ASTMH) and an expert panel of the Infectious Diseases Society of America (IDSA) [43].

In this study, we used EITB assay as the confirmatory test for NCC in conjunction with clinical presentation and neuroimaging data. A total of 150 NCC cases were detected among a total of 970 suspected cases by immunoblot assay, which were referred to PRL, Faculty of Medicine, Kuwait University for disease confirmation during 2006–2019. These cases were detected among 98 expatriates mostly originating from taeniasis-endemic countries, but also, surprisingly, among 52 Kuwaiti nationals. Although largely considered to be taeniasis/cysticercosis-free, a total of 751 patients were also reported from different countries of Western Europe during 1985 to 2011, and the disease was even more common in Spain and Portugal [14,39]. A large case series of neurocysticercosis were recently reported from New York City. The NCC was detected among 260 cases, of which 245 (94.2%) patients were immigrants from 22 different, mostly taeniasis-endemic countries [44].

NCC cases were detected in a relatively younger age group (24.3 years) in Kuwait than the noninfected subjects were (33.1 years; *p* < 0.001); however, gender distribution was similar. Similar results were reported from India [45] and Peru [16], where the mean age of NCC patients was also below 30 years of age. On the other hand, studies from less endemic countries, including the USA, China, Spain, and Indonesia, reported a higher mean age of NCC cases ranging from 37 to 40 years [44,46,47,48].

The majority (98 of 150, 65.3%) of NCC patients in Kuwait were expatriates. Although the country of origin of 20 subjects was not known, the most common countries of origin for the remaining expatriates were India (55 of 150, 36.7%) and Nepal (12 of 150, 8%), which are taeniasis-endemic countries [4,7,45]. Although data on the exact time of entry of these expatriate NCC patients in Kuwait were not available, their majority had been residing in Kuwait for several years and had not travelled to their country of origin for nearly 4–5 years before their presentation. The clinical presentation of patients with NCC is highly variable and nonspecific, and depends on lesion number, site, and size, and the degree of the host’s inflammatory response [49,50]. It is well-documented that infected people remain asymptomatic for many years before presenting with NCC symptoms [1,3,4]. Our results were also consistent with recently reported data from a large case series of NCC patients from New York City. Median time from immigration to disease presentation in this study involving 260 NCC cases was also long (9.5 years) [44].

Although the positivity rate for NCC among suspected cases showed a biphasic curve, disease prevalence generally showed a declining trend in the country as the total population of Kuwait steadily increased from 2.37 million in 2006 to 4.46 million in 2020 [24]. The larger-than-expected number of positive cases detected in 2014–2016 was mainly due to the diagnosis of several NCC cases among the same family members in a few Kuwaiti households, including both Kuwaiti nationals and expatriate domestic servants (described in more detail below). In a recent study, Trevisan et al. [21] reported that nearly 17% of human cysticercosis cases in Eastern Europe were indigenously acquired, even though the source of infection in the majority (69%) of cases was not specified, and a declining trend in disease prevalence was observed, with some countries reporting no new cases in recent years. On the other hand, human cysticercosis cases in Western Europe, a region with marked differences in disease prevalence, were mostly seen among immigrants or returning travelers from endemic countries and were increasing, likely due to increasing immigration of subjects from Latin American and Caribbean countries to Western Europe, particularly southern European (Spain and Portugal) countries [17]. Some human cysticercosis cases in Western Europe also originated from eastern European countries, where more favorable conditions for local *T. solium* transmission exist [17,51]. NCC cases were also previously described from Kuwait and other nearby countries in the Arabian Peninsula [25,32,33,34,35,36,52]. The occurrence of human taeniasis was also recently described in 18 of 21 countries in the Middle East and the North Africa (MENA) region [31].

Due to the retrospective nature of the study, clinical details for only 48 of 50 NCC patients detected during 2014 to 2019 could be retrieved from the medical records. Cysticercal lesions were observed in CT/MRI images in 39 of 48 NCC cases, with 15 cases showing a single lesion, and 24 cases exhibiting multiple brain lesions at multiple sites. The majority of cases showed parenchymal lesions, while subarachnoid lesions were also observed in some cases, and most patients presented with seizures or persistent headache. There is still a lot of debate on the classification of NCC lesions. Most current clinical and radiological studies consider subarachnoid NCC to be the most common form, followed by parenchymal NCC [53]. However, on the basis of histopathologic analysis, some authors argue that parenchymal NCC represents subarachnoid cysticercosis located in deep sulci or in perforating branches of perivascular spaces [54]. In this study, we documented the anatomical site or region of cystic lesions. We recorded 47 cyst locations by neuroimaging, and 59% of cases showed lesions in their frontal lobes, followed by 30.7% in the occipital lobes. Several studies and case reports from other nearby countries also described NCC cases, mostly among young women who had locally acquired the infection and presented with parenchymal brain cysticercus granuloma with minimal or no symptoms [25,32,33,34,35,36]. In a recent study based on a large number of NCC case series from New York City, Berto et al. [44] also reported parenchymal NCC in the majority (139 of 260; 53.5%) of patients, while 40 (15.4%) patients were diagnosed with subarachnoid NCC, and seizures or headache were also the most common symptoms among NCC patients.

NCC cases (*n* = 52) among Kuwaiti nationals were detected among individuals who had never consumed pork, and most of them had also not travelled to any of the countries in which *T. solium* infection is endemic. Kuwaiti people generally like to consume a lot of meat (chicken, lamb, beef, camel, and seafood), dairy products, and sweet dishes. Pork farming is strictly prohibited, and pork is not consumed at all due to the Islamic culture. The total population of Kuwait in 2019 included ~1.3 million citizens who employed >650,000 domestic workers and babysitters mainly originating from India, Sri Lanka, Nepal, the Philippines, and Ethiopia, which are endemic for *T. solium* taeniasis [3,4,36,55] and could, therefore, pose a possible risk of infection transmission to the local Kuwaiti population through the fecal–oral route. This is also supported by a recent study that screened 500 newly arrived domestic workers and found 4.8% of these workers to have high levels of anti-*T. solium* taeniasis-specific IgG antibodies [28]. It was, therefore, reasonable to assume that the source of infection in Kuwaiti NCC cases were expatriate domestic workers who had come from *T. solium*-taeniasis endemic countries. To explore this further, all traceable expatriate domestic workers living in the family homes of Kuwaiti NCC-positive cases during the last few years of the study were also screened. A cluster of 3 families with multiple members detected with NCC since 2014 are noteworthy. This comprised 2 families with 3 members each (one family of 2 sisters and 1 brother of 6–9 years detected in 2014, and another family of 2 sisters and 1 brother of 6–11 years detected in 2016) who had domestic workers from India, Nepal, or the Philippines. Unfortunately, due to the retrospective nature of the study, the expatriate domestic workers living with these two families could not be screened, as they had left their households and were not traceable. However, another family detected in 2014 included 6 NCC-positive households, 4 Kuwaiti members of a family, 2 adult females of 26 and 46 years and 2 young children of 9 and 11 years, and 2 female domestic workers from Sri Lanka and Philippines (Figure 1a). While two of the Kuwaiti subjects were asymptomatic, two other subjects had seizures, and the 46-year-old Kuwaiti female also showed 2 calcified lesions in her CT scan (Figure 1b). In this cluster, the 2 domestic workers living in the same household were available for screening and were strongly positive for *T. solium* taeniasis-specific antibodies to taeniasis-specific rES33 antigen by ELISA [28] and by EITB assay (Figure 1a). Most of the NCC cases among the Kuwaiti nationals were autochthonous, as they occurred in wealthy families who had employed domestic workers from disease-endemic areas. It is likely that some were *Taenia* spp. carriers who transmitted the infection through the nonhygienic handling of food or directly by the fecal–oral route. The source of infection, *Taenia* spp. eggs, was, however, not identified in most of these workers by the microscopy of their stool specimens, as they had either left their workplace and so were not available for screening or due to the poor sensitivity of the microscopy. Several studies from Europe, New York, and other nonendemic countries reported similar findings [11,18,20,44,52].

Only limited treatment information was available for the 48 NCC cases detected during 2014–2019. Symptomatic therapy was the focus of the initial and emergency management of our patients; thus, antiepileptic drugs were used to control seizures. All patients were admitted before starting antiparasitic drugs. Patients with viable parenchymal cysticerci were mostly treated with albendazole (15 mg/kg/day), alone or combined with praziquantel (50 mg/kg/day). The patients showed better outcome when treated with antiparasitic drugs along with corticosteroids. Almost all NCC cases had a favorable clinical outcome, though some cases required prolonged treatment of >6 months, followed up clinically and by neuroimaging. The two domestic workers who were implicated as the source of infection in one Kuwaiti family showed no apparent signs of active infection (worm segments in stool samples) during treatment. However, this observation has also been noted in few other studies [4,13,56].

Our study has a few limitations. The EITB assay was performed only on those individuals who were suspected for NCC on the basis of imaging data by MRI or CT scans. Demographic, clinical, and other details were not available for all EITB-positive NCC cases due to the retrospective nature of the study. Serum samples from domestic workers employed by some EITB-positive Kuwaiti cluster cases were not available, as they had left their employers.

## 5. Conclusions

This is the first study reporting both the diagnostic and clinical characteristics of a large series of NCC patients detected from 2006 to 2019 in Kuwait. A total of 150 of 970 (15.5%) suspected cases were positive by EITB, and comprised 98 expatriates mostly originating from taeniasis-endemic countries, but also, surprisingly, 52 Kuwaiti nationals with no history of pork consumption or travel to taeniasis-endemic countries. Cysticercal lesions in various brain regions were found in 39 of 48 patients diagnosed during 2014–2019, and seizures and persistent headache with/without fever were the most common symptoms. Multiple members of 3 families with NCC were identified; infection was linked to domestic workers from taeniasis-endemic countries and confirmed in at least 1 family. Our data show that NCC is predominantly imported in Kuwait by expatriates originating from taeniasis-endemic countries who transmit the infection to Kuwaiti citizens. The data also suggest that domestic workers and food handlers who likely transmitted the infection were not identified during the screening of their stool samples, possibly due to the lower sensitivity of the employed procedure (microscopy). It is, therefore, recommended that all domestic workers and food handlers also be screened by the more sensitive rES33 antigen ELISA in addition to stool microscopy, and appropriately deworm all who show anti-*T. solium* taeniasis-specific IgG antibodies to prevent the transmission of infection to the local population.

## Figures and Tables

**Figure 1 microorganisms-09-01221-f001:**
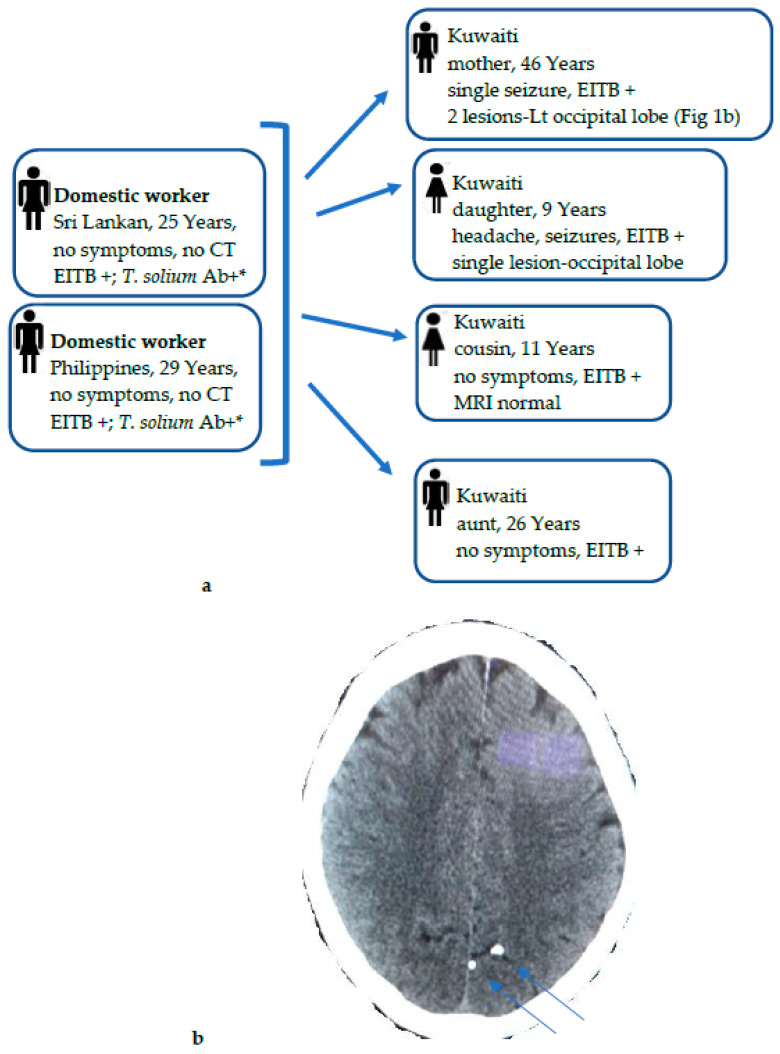
(**a**) Diagnostic workout and contact tracing of cysticercosis infection among 6 NCC households, 4 Kuwaiti family members (mother, daughter, aunt, and cousin), and 2 domestic workers, all living in the same house. The possible source of *T. solium* infection was linked to 2 domestic workers from Sri Lanka and Philippines, who showed high levels of anti-*T. solium* taeniasis-specific IgG antibodies detected by ELISA using *T. solium*-specific rES33 antigen *. (**b**) CT scan of Kuwaiti mother with NCC who presented with a single tonic clonic seizure. Her CT scan showed 2 calcified lesions on the left occipital lobe (marked by arrows), with no surrounding edema. She was also positive for cysticercus-specific IgG antibodies by EITB.

**Table 1 microorganisms-09-01221-t001:** Number of serum samples screened for cysticercosis and NCC and immunoblot positivity data reported in this study.

Variable	Total	Immunoblot	Immunoblot	*p* Value
	Subjects	Negative	Positive	
Mean age (years)	31.8	32.8	24.3	0.001
Gender				
Male	482	409	73	0.913
Female	488	411	77	
Country of origin				
Kuwait	325	273	52	
India	287	232	55	
Nepal	67	55	12	
Bangladesh	51	47	4	
Syria	30	27	3	
Philippines	33	32	1	
Sri Lanka	29	28	1	
Pakistan	15	14	1	
South Korea	9	8	1	
Unknown *	124	104	20	

* Expatriates or individuals whose nationalities were not recorded in the logbooks.

**Table 2 microorganisms-09-01221-t002:** Annual number of NCC cases detected in Kuwait among Kuwaiti nationals and expatriates from different countries during 2006 to 2019.

Year	No. of Screened Subjects	No. of EITB Assay	Nationality of Immunoblot-Positive Neurocysticercosis Cases (*n*)									
		Positive Cases (%)	Kuwaiti	Indian	Nepali	Bangladeshi	Syrian	Filipino	Sri Lankan	Pakistani	South Korean	Unknown
2006	51	12 (23.5)	2	8	1							1
2007	87	12 (13.8)	1	4	1	1	1					4
2008	84	17 (20.2)	4	7	1		2					3
2009	74	19 (25.7)	3	6	2	3						5
2010	47	9 (19.1)	4	4	1							
2011	92	21 (22.8)	8	10								3
2012	83	7 (8.4)	2	4	1							
2013	25	3 (12)	2									1
2014	137	23 (16.8)	16	3	2			1	1			
2015	81	16 (19.7)	6	6	2					1	1	
2016	110	9 (8.2)	4	2								3
2017	25	0 (0)										
2018	43	2 (4.6)		1	1							
2019	31	0 (0)										
**Total**	**970**	**150 (15.5)**	**52**	**55**	**12**	**4**	**3**	**1**	**1**	**1**	**1**	**20**

EITB, enzyme-linked immunoelectrotransfer blot.

**Table 3 microorganisms-09-01221-t003:** Symptoms and clinical presentation among 48 NCC cases detected during 2014–2019.

Symptoms or Clinical	No. (%) of Patients
Presentation	Testing Positive
Tonic–clonic seizures	36 (75%)
Persistent headache	14 (29.2%)
Persistent headache with fever	7 (14.6%)
Seizures with loss of consciousness	6 (12.6%)
Right hemiparesis	4 (8.3%)
Vomiting and hypertension	3 (6.3%)
Right hemiplegia	1 (2.1%)
No symptoms	9 (18.7%)

**Table 4 microorganisms-09-01221-t004:** Site of cysticercal lesions on neuroimaging (MRI/CT scans) among 39 NCC cases detected during 2014–2019.

Location of Cysticercal Lesions	No. (%) of Patients
in MRI/CT Scans	with Lesions
Frontal lobe	3 (7.7%)
Parietal lobe	3 (7.7%)
Occipital lobe	2 (5.1%)
Temporal lobe region	3 (7.7%)
Cerebral hemisphere	1 (2.6%)
Frontal and occipital lobes	3 (7.7%)
Frontal and parietal lobes	4 (10.3%)
Frontal and temporal lobes	3 (7.7%)
Frontal, temporal, and occipital lobes	3 (7.7%)
Frontal, occipital lobes, and temporoparietal region	3 (7.7%)
Right hemisphere and frontal lobe	2 (5.1%)
Right hemisphere and cerebellum	2 (5.1%)
Frontal lobe and pineal body	2 (5.1%)
Occipital lobes and cerebral tonsil	1 (2.6%)
Other sites *	5 (12.8%)

* Other sites included multiple cysticercal lesions in the postcentral gyrus, cerebellum, posterior fossa in combination with other locations, including extra-axial CNS cysts (pterygoid muscles, parotid gland, temporalis muscle).

## Data Availability

All data are available within the manuscript.

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
