# Peer review of "A Large Case Series of Neurocysticercosis in Kuwait, a Nonendemic Arabian Gulf Country in the Middle East Region"

_microorganisms, 2021, doi:10.3390/microorganisms9061221_

Round 1
Reviewer 1 Report
The manuscript entitled “A large case series of neurocysticercosis in Kuwait, a non-endemic Arabian Gulf Country in the Middle East Region” is well structured and written by the Authors.
What is especially relevant is the epidemiological analysis that emphasizes the importance of parasite transmission in endemic areas.
I do not have any observation to put to the Authors regarding the experimental part, even if I would like, if possible, that they would report also the values of specificity as well as sensitivity of EITB employed in the survey.
It would also be extremely interesting, also for a retrospective analysis of the risk attributable to different factors, to have the medical records of the 970 patients. I realize it is very difficult, but at least some aspects related to eating habits and some other aspects that can stratify the population of subjects not only based on travel. If it were possible to find these cards, the authors could collect this information in a table with relative analysis of attributable risks to be included as "supplementary material".
Apart from this suggestion, which is obviously not binding, but it could only give more value to the investigation of itself very valid, I renew my compliments to the Authors for the work done and the scientific rigor shown.
Reviewer 2 Report
The MS by Iqbal and cols. analyzed the occurrence of NCC in Kuwait in a retrospective study based on samples received in the Parasitology Reference Laboratory according to a history leading to NCC suspicion. It is an informative study, however, I may point several issues to be improved in a possible revised version as follows:
-What is the origin of the samples? is it possible to determine by area of the country, as the samples were from different hospitals? Do the authors think the found numbers can express a real prevalence of the disease?
-Is not a countrywide sampling, the title should be modified.
-The MS is mixing sections eg. in the methods there is a long discussion, which should be transferred for the appropriate section of the MS.
-Is it possible to determine where the Kuwaiti subjects acquired the infection (other than the 2 traced cases)?
-If possible describe the resolution of the treated cases.
-Is there any recommendation that could be done based on the results of this study?
-The conclusion could be more concise.
Round 2
Reviewer 2 Report
All the suggestions and comments were satisfactorily addressed.